# Urban Land Intensive Use Evaluation Study Based on Nighttime Light—A Case Study of the Yangtze River Economic Belt

**Xin Cheng** [1]**, Hua Shao** [1,]*****, Yang Li** [2,3,4]**, Chao Shen** [1] **and Peipei Liang** [1]

[1] College of Geomatics Science and Technology, Nanjing Tech University, Nanjing 211816, China; chengxin2016@njtech.edu.cn (X.C.); sc1015147151@njtech.edu.cn (C.S.); 201861223009@njtech.edu.cn (P.L.)

[2] Jiangsu Center for Collaborative Innovation in Geographical Information Resource Development and Application, Nanjing Normal University, Nanjing 210023, China; li.yang@njnu.edu.cn

[3] Key Laboratory of Virtual Geographic Environment, Nanjing Normal University, Ministry of Education, Nanjing 210023, China

[4] School of Geographic Science, Nanjing Normal University, Nanjing 210046, China

***** Correspondence: shaohua@njtech.edu.cn; Tel.: +86-025-5813-9842

**Abstract:** Urban land intensive use meets the requirements for the sustainable development of urban land and is an important part of urban sustainable development. The Yangtze River Economic Belt (YREB) spans the three major regions of China, which are the most active areas of China's economy. The contradiction between humans and land is becoming more acute. There are also regional differences in land use patterns affecting the coordinated development of the YREB and the construction of an ecological civilization. Therefore, the scientific evaluation of urban land intensive use is a key area in the current research field of urban sustainable development. In this study, the YREB is chosen as the research object, and urban land intensive use is studied using nighttime light data and statistical data on the urban built-up area. An evaluation model based on urban nighttime light intensity and land urbanization is constructed with an allometric growth model. Considering that the impact of land urbanization on urban nighttime light has a possible lag effect, an evaluation model of land intensive use that considers the lag effect between urban nighttime light and the land urbanization level is proposed. Using urban agglomerations and some typical cities in the study area as research samples, the characteristics of urban nighttime light and land urbanization are analyzed to reveal the spatial and temporal characteristics of land development in the YREB. The results show that nighttime light remote-sensing data can better reflect the level of urban land use, the allometric growth model can better fit the intensity of urban light and the land urbanization level, and the allometric growth characteristics can reflect the land use characteristics of different cities and urban agglomerations. In regional experiments with typical cities and with urban agglomerations, compared to the original allometric growth model, the goodness of fit of the allometric growth model with the lag effect improves, on average, by 3.2% and 2%, respectively, with the highest increases being by 9.9% and 4.9%, respectively. The level of intensive land use in the YREB gradually decreases from east to west, and there are great differences among different cities in the provinces and urban agglomerations. The lower reaches of the Yangtze River have high land intensive use on the whole. In the middle reaches, multicenter cities have a greater efficiency of land use than the surrounding cities. In the upper reaches, only Chengdu and Chongqing have clear advantages in urban land intensive use. The results of this study can be helpful in providing an important reference for the sustainable development of land in the YREB and can provide a basis for future urban land optimization and sustainable development. Realizing the coordination and linkage between key cities and major cities is the key to enhancing the overall sustainable development ability of the core cities in the YREB.

**Keywords:** land intensive use; remote sensing; nighttime light; allometric growth; Yangtze River Economic Belt; land utilization

---

## 1. Introduction

The Yangtze River Economic Belt (YREB) is a new support belt for China's economy in the form of an urban agglomeration proposed based on the Golden Channel of the Yangtze River [1]. In 2016, the Outline of the Development Planning of the YREB established a new economic development pattern of "one axis, two wings, three poles and multiple points" in the YREB. The "one axis" is the golden waterway of the Yangtze River. The "two wings" refer to the two major transport channels of Shanghai-Chengdu and Shanghai-Ruijin, and the "three poles" refer to the Yangtze River Delta urban agglomeration, the middle reaches of the Yangtze River urban agglomeration and the Chengdu-Chongqing urban agglomeration. Finally, the "multiple points" refer to cities outside the three major urban agglomerations. Previous studies and discussions of the Yangtze River Basin have mainly focused on the Yangtze River Delta urban agglomeration. With economic development entering a new stage, inland economic development will play an important role in national economic development. The YREB has become one of the most powerful and strategic support regions in China. Land resources are important natural resources and provide materials for production in the process of economic development. Therefore, human beings must improve the efficiency of land utilization and realize the sustainable utilization of land resources in the process of land development [2]. In the process of urbanization, the area of urban construction is gradually expanding, the benefits of urban construction land are constantly improved, and a large number of people are gathered in cities. However, due to regional economic development and the chaotic expansion of cities and towns, the utilization of urban land resources is inefficient [3]. The need to manage urban sprawl and its manifold adverse consequences by promoting compact urban development should be promoted in science and policy-making [4]. Evaluating and analyzing the intensity level of construction land are helpful for understanding the spatial differentiation and change trends in urban land intensive use and provide a basis for future urban land optimization and sustainable development [5]. Studying land intensive use at the scales of the cities, provinces and urban agglomerations in the YREB is of great significance for understanding the land use situation there, avoiding the rapid and chaotic expansion of cities and preserving the value of land for construction and the rational utilization of land resources.

In our research, we focus on urban land use efficiency. If limited developed urban land produces considerable economic benefits, then the urban land use efficiency is high, which means that it is intensive. Urban land use efficiency can be measured by population density, the volume rate and the rate of land output in cities [6]. Traditional research on urban land intensive use mainly involves related theories and their implications, the driving forces and influencing factors and the evaluation of urban land intensive use [7]. The analytic hierarchy process (AHP) is the most common method in evaluating land intensive use; it requires the collection and collation of various types of socioeconomic statistics, constructs an evaluation index and determines the weight of each index [8,9]. Although the method is simple and operable, subjective judgment and preference greatly affect the results [10]. In addition, multifactor comprehensive evaluation [11,12], principal component analysis [13,14], global principal component analysis [15], the back propagation (BP) neural network model [16], clustering analysis [17], the construction of evaluation frameworks with spatially explicit information [18] and other methods have gradually been applied as methods of evaluating urban land intensive use. However, these methods require the support of a large number of socioeconomic statistics, and there are many problems, such as difficulties in collecting statistics, time-consuming basic data processing, data "monopoly" and data distortion. With the development of remote-sensing technology, the advantages of remote-sensing images for earth observation have allowed scholars to begin applying remote-sensing data to research land intensive use [19,20]. Remote-sensing data

have incomparable advantages over traditional statistical data in terms of spatiotemporal resolution, accessibility and accuracy.

In 1976, the F10 satellite launched by the US National Defense Meteorological Satellite Program (DMSP) carried operational linescan system (OLS) sensors for the first time. Subsequently, the F12 (1994–1999), F14 (1997–2003), F15 (2000–2007), F16 (2004–2009), and F18 (2010–2013) satellites were successively launched. These satellites could obtain global nighttime light brightness data and provide the longest time series (1992–2013) of nighttime light image data. Deren Li [21] pointed out that nighttime light has the unique ability to reflect human social activities. At the same time, the space-time continuity of nighttime light data can effectively compensate for the shortcomings of incomplete traditional counting data and statistical gaps, and the processing of such data is more convenient. Currently, nighttime light data are widely used in urban spatial data mining, for example, in population estimation [22–24], GDP estimation [25,26], urban expansion monitoring [27–29], urban economic efficiency evaluation [30,31], the spatial characterization of urban development [32–34], and the spatiotemporal characterization of urbanization dynamics [35,36]. Various research results have shown that nighttime light correlates well with the various socioeconomic indicators of a city and can reflect the degree of economic activity in a city. However, few studies have assessed intensive land use based on the relationship between the growth in nighttime light intensity and the expansion of urban land. In contrast, the allometric growth model has been widely used to study urban geography. Allometric growth analysis is of great significance for research on the scaling characteristics among the elements of urban systems [37,38]. Analyzing the relationship between an urban population and an urban area, Yanguang Chen et al. [39] found that they obey the law of allometric growth at the municipal level. The law of allometric growth provides an important scientific basis for coordinating the relationship between humans and land in the process of urbanization [40,41]. However, previous studies of allometric growth have mostly discussed the relationship and mechanism of development between population and land urbanization. In this paper, we consider that there is a certain relationship between the growth in nighttime light intensity and the urban land area; consequently, nighttime light and the urban land area are innovatively used to construct an allometric growth model. The built-up area is an important index reflecting the land urbanization level, and nighttime light data can reflect the vitality of urban development. Therefore, the degree and efficiency of urban land use can be understood by constructing the allometric growth relationship between nighttime light and the urban built-up area.

In light of the above considerations, this paper aims to study the intensive land use level in the YREB. We attempt to introduce nighttime light data and statistical data on the built-up area to construct a model of urban land intensive use based on the allometric growth model to spatially analyze the level of urban land intensive use. Then, the model is improved by considering the lag effect between nighttime light and urban expansion to further determine the characteristics of land intensive use in the YREB. The evaluation of urban land intensive use in the YREB is helpful for understanding the efficiency of urban land during urban expansion and provides a basis for urbanization in the future. The feasibility of intensive utilization provides a reference for land use efficiency evaluations at different spatial scales within the YREB.

## 2. Study Area and Data Sources

### 2.1. Study Area

The YREB (Figure 1), which extends from Shanghai in the east to Yunnan in the west, covers the nine provinces and two cities of Shanghai, Chongqing, Jiangsu, Hubei, Zhejiang, Sichuan, Yunnan, Guizhou, Hunan, Jiangxi, and Anhui. The economic belt spans the three major regions of eastern, central and western west China and runs through the hinterland of Central China. Its population and GDP account for more than two-fifths of those of the entire country. It is the most developed region in the Yangtze River Basin and one of the highest-density economic corridors in China. Three of



the five national-level urban agglomerations identified in the National New Urbanization Plan are located in the YREB, namely, the Yangtze River Delta urban agglomeration, the middle reaches of the Yangtze River urban agglomeration and the Chengdu-Chongqing urban agglomeration. The overall topography of the YREB gradually decreases from the west to the east, and the topography of the basin is complex, including plateaus, mountains, basins, hills, and plains. Furthermore, the land use structure and the degree of development and utilization are characterized by diversity [42]. Influenced by natural conditions, geographical location, historical development and regional policies, the distribution of urbanization in the economic belt is imbalanced. The population in the urban area of the YREB has clear "circle characteristics" [43].

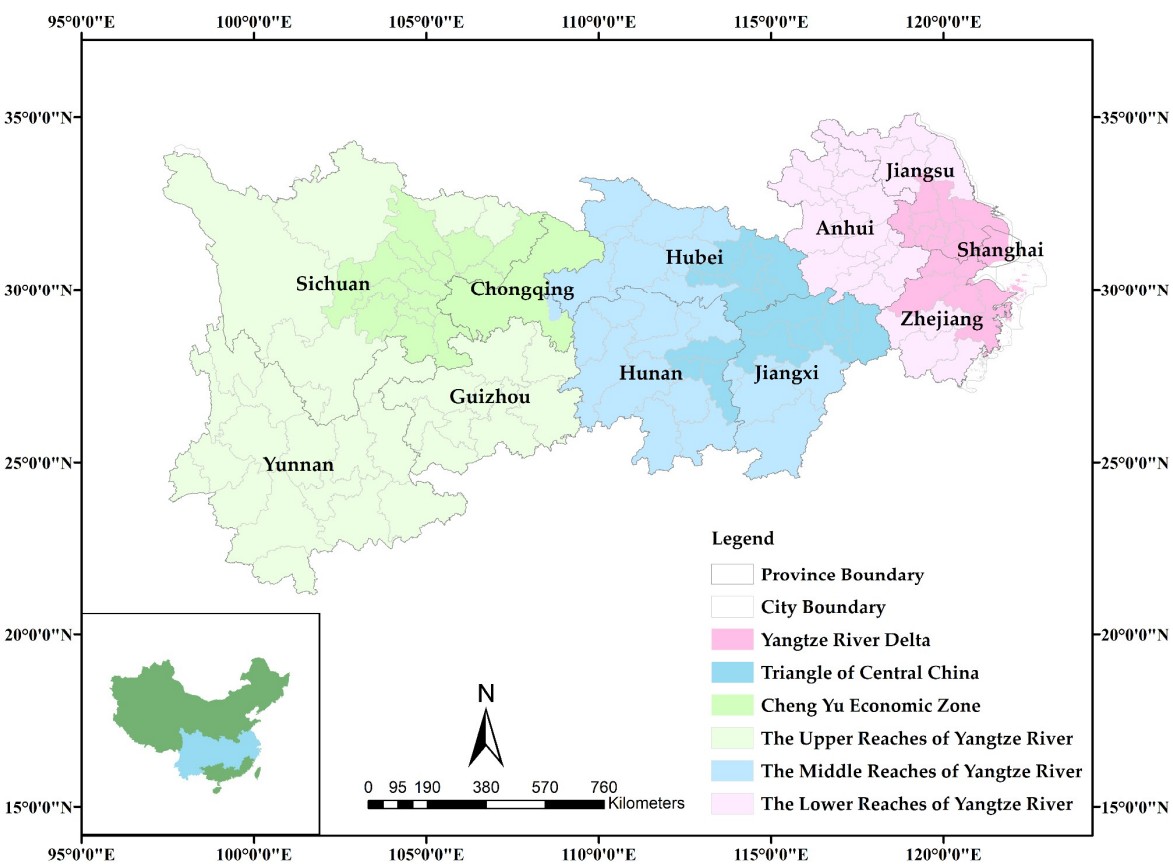

**Figure 1.** Study area of the Yangtze River Economic Belt (YREB).

*2.2. Data Sources*

The DMSP-mounted OLS sensor provides the world's longest time series (1992–2013) of nighttime light image data. DMSP/OLS nighttime stable light data (NSL) are annual raster images for calibrating average night light intensity. These images include persistent light sources in towns and other places, and they remove moonlight, clouds, fires, and the burning of oil and gas. The image reference frame is the WGS-84 coordinate system. The width is 3000 km, the spatial resolution is approximately 1 km, and the range of the image digital numbers (DNs) is 0–63. NSL data are currently the most commonly used source of nighttime light remote-sensing data. However, these data have certain shortcomings, such as low resolution, oversaturation in the center of cities [44], and a halo at the boundary of cities [45]. Therefore, a series of pretreatments should be performed before using NSL data to ensure the accuracy of the test results.

The data selected for this study include DSMP/OLS NSL data for 2001–2013 from the website of the National Geographic Data Center of the National Oceanic Atmospheric Administration, statistical data on the built-up areas of cities at all levels in the YREB for 2001–2013, which come from the

statistical yearbook on the official website of the Ministry of Housing and Urban-Rural Development of the People's Republic of China, and shapefiles of the administrative divisions of the YREB from the Yangtze River Delta Science Data Center, National Earth System Science Data Sharing Infrastructure, National Science and Technology Infrastructure of China.

## 3. Methodology

### 3.1. Preprocessing of DMSP/OLS Nighttime Light Data

To study the intensive land use in the YREB from these perspectives, the degree of economic activity and the land urbanization level are the variables in this study. The average nighttime light intensity obtained from DMSP/OLS NSL data is used to represent the degree of economic activity [21]. The statistical data on the built-up areas of cities are used to calculate the average built-up area in each city to represent the land urbanization level in the study area. Then, the allometric model is utilized to demonstrate the allometric growth relationship between nighttime light intensity and the built-up areas. When DMSP/OLS nighttime light data are used in research, the main problems are as follows [46]: 1) there are differences between images captured by sensors in the same year; 2) the DN values of pixels fluctuate abnormally in the same location of images in different years; and 3) there is oversaturation. To ensure the accuracy of the research results, we preprocessed the nighttime light data by correcting invariant target regions [47] and calibrating the DMSP/OLS nighttime light data (2001–2013) with the radiometric calibrated F162006 images provided by the NOAA [48] and same- and different-year image correction between sensors. The final correction results are shown in Figure 2.

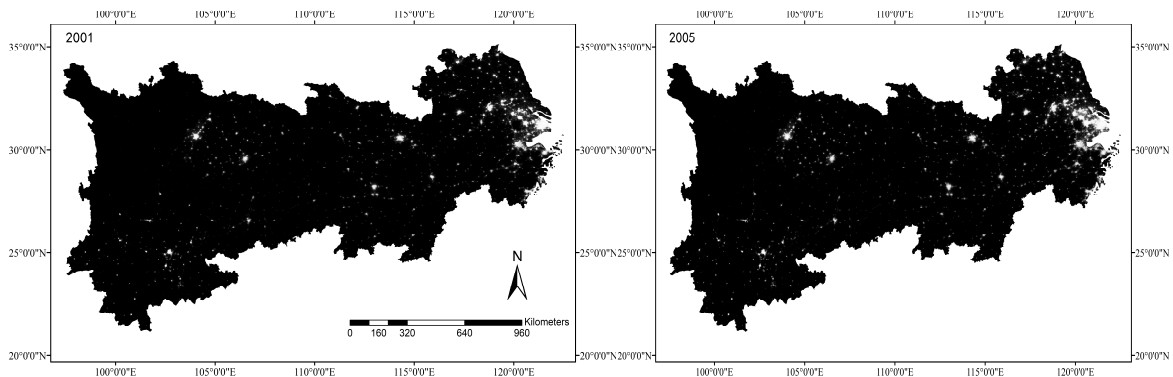

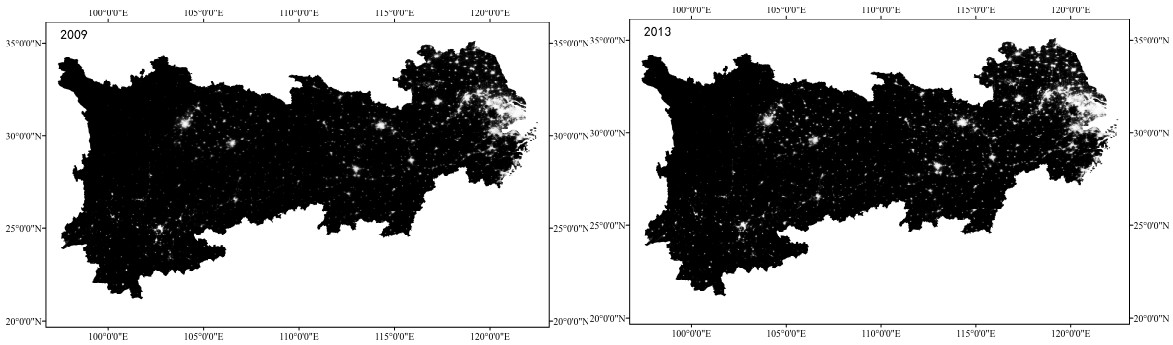

**Figure 2.** YREB nighttime stable light (NSL) data after correction.

### 3.2. Allometric Growth Model of Nighttime Light Intensity and the Land Urbanization Level

The law of allometric growth was first proposed in biology and ecology [49]; subsequently, it was introduced into urban geography to study scale characteristics among the elements of an urban system

or between an element and the whole system. This method was later used to describe the nonlinear relationship between urban populations and urban areas in research on urbanization.

Allometric growth refers to the geometric measure relationship between one part and the whole system or another part of the system [50]. The relationship can be expressed as Equation (1):

$$\frac{1}{y}\frac{dy}{dt} = b\frac{1}{x}\frac{dx}{dt} \qquad (1)$$

where y is a local element, x is a system or another local element, t is time, and b is the allometric growth coefficient or the ratio of the growth rate between x and y. When the two correlation measures of a system satisfy Equation (1), the system obeys the law of allometric growth. Then, the formula can be transformed into a power function:

$$y = ax^b \qquad (2)$$

Theoretically, if the two elements of the system satisfy the geometric measure relationship, the two elements must have the characteristic of allometric growth [37]. In the logarithmic transformation of Equation (2), the model of the allometric growth in nighttime light intensity and the land urbanization level is as follows:

$$\ln y = \alpha_0 + \alpha_1 \ln x \qquad (3)$$

where $\alpha_0$ is the constant term, and $\alpha_1$ is the allometric growth coefficient.

Since nighttime light has the unique ability to reflect human social activities and economic development, it can be regarded as an element of urban systems. Urbanization enables and influences the growth in nighttime light, which can reflect whether the developed land is utilized efficiently. Therefore, there can be a certain relationship between their growth rates. We use the construction method of the allometric growth model for reference to construct the allometric growth model of nighttime light intensity and the land urbanization level. Based on the allometric growth coefficient, the urban land intensive use levels in different cities or areas are obtained.

The allometric growth model of nighttime light intensity and the land urbanization level is constructed as follows:

$$\ln L = \alpha_0 + \alpha_1 \ln S \qquad (4)$$

where $\alpha_0$ is the constant term and $\alpha_1$ is the regression coefficient. $L$ is the average nighttime light intensity per square kilometer within the administrative divisions, and $S$ is the land urbanization level in the n-year administrative region, which is 100 times the average built-up area per square kilometer.

According to the regression coefficient, we can obtain the allometric growth characteristic between nighttime light and land urbanization. When $\alpha_1$ is greater than 1, the growth rate is positively allometric; the increased rate of nighttime light intensity is greater than the expansion rate of urban land, indicating that the land use is intensive. When $\alpha_1$ is equal to 1, the growth rate is isometrically allometric, which means that the growth rate of nighttime light is nearly the same as the expansion rate of urban land. When $\alpha_1$ is less than 1, the growth rate is negatively allometric; the growth rate of nighttime light intensity is slower than the expansion rate of urban land, indicating that the urban land is extensive. Therefore, land use efficiency is estimated by the type of allometric growth; when the allometric growth is positive or isometric, the land use efficiency in cities is good.

### 3.3. Allometric Growth Model Considering the Nighttime Light Lag Effect

In the allometric model proposed above, the nighttime light intensity of each year corresponds to only the value of the urbanization level in the same year. We considered that it takes time for new land to increase the nighttime light intensity, which means that the impact of land urbanization on urban light intensity has a certain lag effect. Drawing on related research [41], the previous land urbanization level is introduced into the allometric growth model to determine the contributions of new land and stock land on urban nighttime light, which helps analyze the land use pattern during the expansion

of urban land. We use the current urban nighttime light intensity as the explained variable and the previous land urbanization level as the explanatory variable to construct the following model:

$$\ln L_n = \beta_0 + \beta_1 \ln S_n + \beta_2 \ln S_{n-1} \tag{5}$$

where $\beta_0$ is the constant term, $\beta_1$ is the allometric growth coefficient of the current period, $\beta_2$ is the lag term coefficient, $L_n$ is the average nighttime light intensity of cities in the nth year, $S_n$ is the land urbanization level in the *n*th year, and $S_{n-1}$ is the land urbanization level in the *(n–1)*th year.

Unlike the previous model, this model contains two coefficients to explain the level of urban land intensive use. When $\beta_1$ is greater than $\beta_2$, the land use efficiency of new land is higher than that of developed land; when $\beta_1$ is less than $\beta_2$, the land use efficiency of new land is lower than that of developed land. This method is specifically designed to analyze land use efficiency in a time series.

## 4. Results and Analysis

*4.1. Results of the Evaluation Model of Urban Land Intensive Use in the YREB*

### 4.1.1. Results of the Allometric Growth Model Based on Nighttime Light and the Land Urbanization Level

This paper uses the allometric growth model to reflect the growth relationship between urban nighttime light intensity and the land urbanization level. The nighttime light intensity is the explained variable, and the land urbanization level is the explanatory variable. Based on order space, the allometric growth coefficients of the urban average light intensity and the land urbanization level of 110 prefecture-level cities in the YREB from 2001 to 2013 are analyzed. Natural fracture point classification [34] is used to classify the allometric growth coefficients of each city. In addition, the distribution of the allometric growth coefficients in the upper, middle and lower reaches of urban agglomerations is compared and analyzed (Figures 3 and 4).

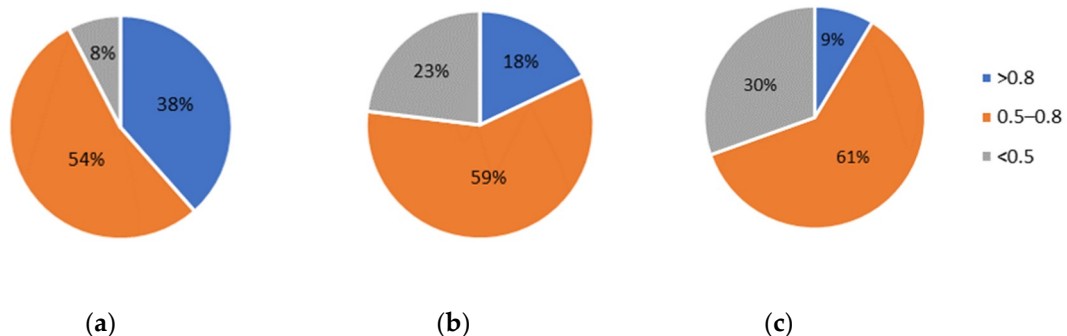

(**a**)　　　　　　　　　　　(**b**)　　　　　　　　　　　(**c**)

**Figure 3.** Comparison of the distributions of allometric coefficients in urban agglomerations, with (**a**) the distribution of allometric coefficients in the lower reaches, (**b**) the distribution of allometric coefficients in the middle reaches, and (**c**) the distribution of allometric coefficients in the upper reaches.

Among the urban agglomerations in the lower, middle and upper reaches of the Yangtze River, the cities with allometric growth coefficients greater than 0.8 accounted for 39%, 18%, and 9%, respectively; the cities with allometric growth coefficients between 0.5 and 0.8 accounted for 54%, 59%, and 65%, respectively; and the cities with allometric growth coefficients less than 0.5 accounted for 7%, 23%, and 26%, respectively. The proportion of cities with higher levels of land intensive use in the lower reaches of the Yangtze River is much higher than those in the middle and upper reaches of the Yangtze River. Additionally, the proportion of cities with moderate degrees of land intensive use in the upper and middle reaches of the Yangtze River is equal, and the proportion in the lower reaches is higher. Figure 4 shows that the allometric growth coefficients of Shanghai, Suzhou, Wuxi and Nanjing in the lower reaches of the Yangtze River are clearly greater than 1; additionally, the allometric

growth coefficients of Changzhou, Taizhou, Hangzhou, Huzhou, Ningbo, and Wenzhou are close to 1. The allometric growth coefficients of Wuhan, Jingzhou, and Changsha in the middle reaches of the Yangtze River are lower than those in the eastern developed areas, but they are clearly superior to those in the surrounding cities. However, the upper and lower reaches of the Yangtze River are separated by a gap. Only Chengdu and Chongqing have higher allometric growth coefficients. The allometric growth coefficient is generally lower in the upper reaches of the Yangtze River.

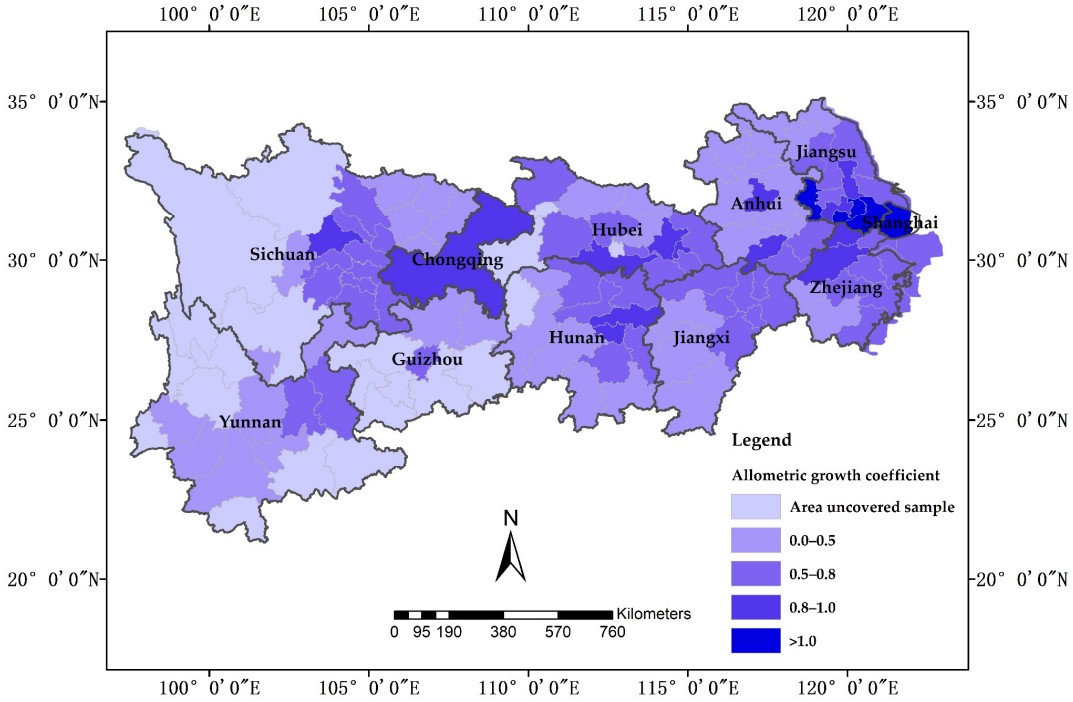

**Figure 4.** Distribution map of the growth coefficients in the YREB.

Based on the model results above, to compare the level of land intensive use among the urban agglomerations in the YREB, this paper selects three national-level urban agglomerations in the YREB: the Yangtze River Delta urban agglomeration; the middle reaches of the Yangtze River urban agglomeration; and the Chengdu-Chongqing urban agglomeration. The Yangtze River Delta urban agglomeration includes Shanghai, parts of south-central Jiangsu, parts of Northern Zhejiang and parts of eastern Anhui. Wuhan city center, the Changsha-Zhuzhou-Tan urban agglomeration and the Poyang Lake urban agglomeration are all located in the middle reaches of the Yangtze River urban agglomeration, and together, they constitute the Triangle of Central China. Since this area contains more cities and the degree of urban economic development differs, the three agglomerations in the middle reaches are discussed separately. The Chengyu city group has Chengdu and Chongqing as its core cities and includes some surrounding cities. Based on the allometric model test results of the urban agglomerations in the YREB, the fitting results are shown in Figure 5.

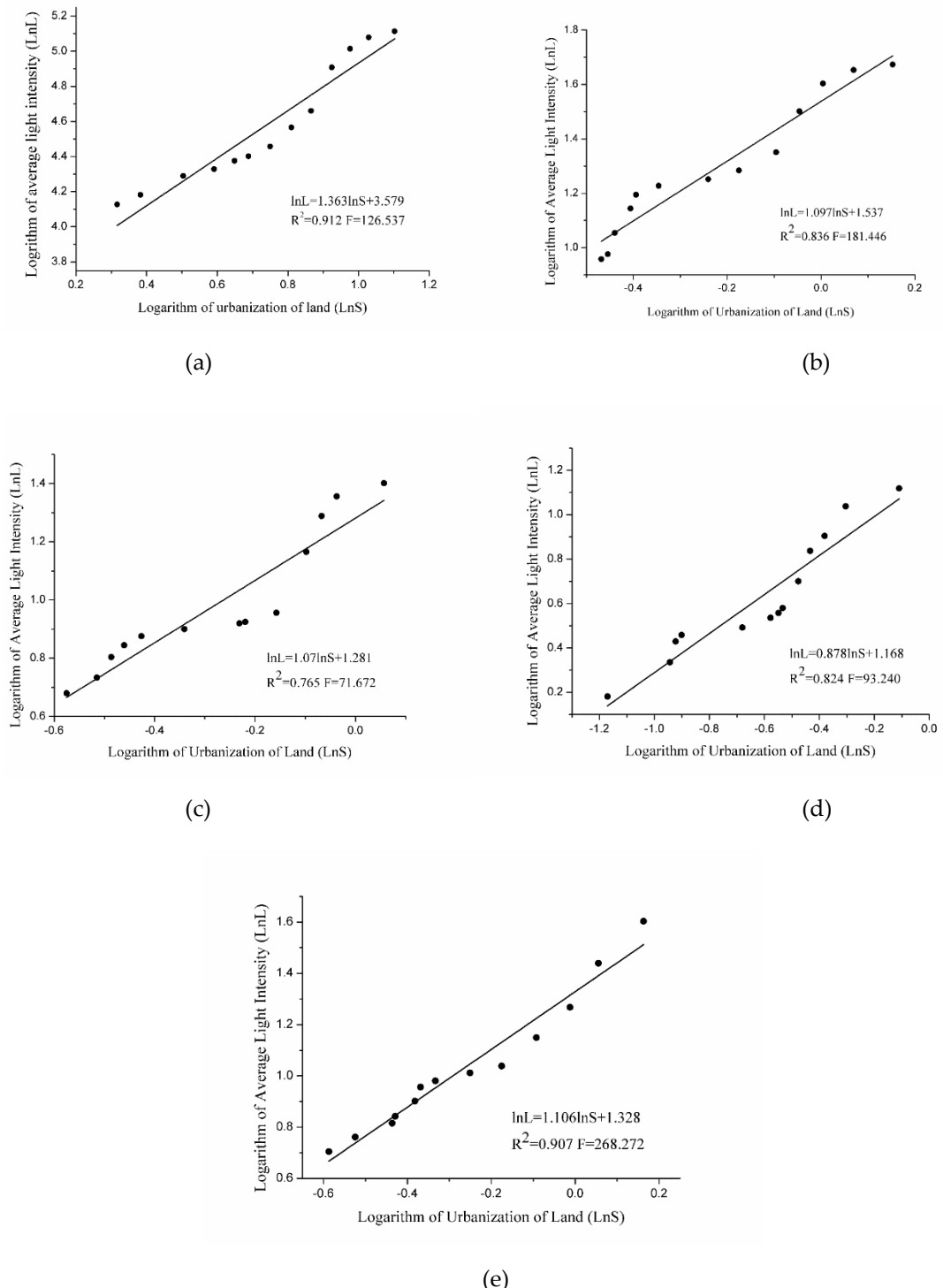

**Figure 5.** Allometric model results of the urban agglomerations in the YREB: (**a**) Yangtze River Delta urban agglomeration, (**b**) Wuhan city circle, (**c**) Changsha-Zhuzhou-Tan urban agglomeration, (**d**) Poyang Lake urban agglomeration, and (**e**) Chengdu-Chongqing urban agglomeration.

The allometric growth model has a good fit with the urban agglomerations in the YREB. The allometric growth coefficient of the urban agglomerations in the Yangtze River Delta is 1.363, which is clearly greater than those of the other four urban agglomerations. This result indicates that the utilization rate of urban construction land in the Yangtze River Delta is high and that economic activity is increasing. The allometric growth coefficients of Wuhan city circle, the Changsha-Zhuzhou-Tan urban agglomeration and the Chengdu-Chongqing urban agglomeration are 1.097, 1.070, and 1.106,

respectively. The allometric growth coefficients of these three urban agglomerations are approximately 1, and the land urbanization expansion and light growth are basically the same. The allometric growth coefficient of the Poyang Lake urban agglomeration is 0.878, and its urban land efficiency is lower than those in the other four urban agglomerations.

### 4.1.2. Results of the Allometric Growth Model Considering the Lag Effect

Because of the lag effect of the nighttime light increase relative to urban land expansion, this paper selects cities with a permanent population of more than 2 million, including Shanghai, Nanjing, Suzhou, Wuxi, Chengdu, and 19 other cities, as a sample to analyze the lag effect. The goodness of fit of the allometric growth model considering the lag effect is clearly improved, indicating a lag effect between nighttime light growth and urban land expansion, and the light intensity is affected by the new land.

Table 1 shows that for most cities, the goodness of fit with the lag term model is better than that of the original allometric growth model. The greatest increase is 9.9%, and the average increase is 3.2%. This result shows a lag effect between lighting and urban land, and the $\beta2$ values of Shanghai, Wuxi, Changzhou, Changsha, Yangzhou, Guiyang, Wuhan, Ningbo, and Wenzhou are all greater than the $\beta1$ values, indicating that the impact of stock construction land on the growth in nighttime light in these cities is greater than the impact of the new land. However, the $\beta1$ values of Nanjing, Suzhou, Hangzhou, Chongqing, Kunming, Nantong, Nanchang, Shaoxing, and Taizhou are all higher than the $\beta2$ values, indicating that the intensity of new land development in these areas is high and that the new land is intensively used. The $\beta1$ and $\beta2$ values of Chengdu and Hefei are similar, and the contribution rate of stock land and new land to nighttime light is balanced.

**Table 1.** Allometric growth model results of typical cities considering the lag effect.

| City | $\alpha1$ | Original Model $R^2$ | $\beta1$ | $\beta2$ | New Model $R^2$ |
|---|---|---|---|---|---|
| Shanghai | 1.25 | 0.702 | 0.4 | 1.01 | 0.823 |
| Nanjing | 1.047 | 0.819 | 0.838 | 0.381 | 0.897 |
| Suzhou | 1.043 | 0.833 | 0.689 | 0.328 | 0.881 |
| Wuxi | 1.013 | 0.909 | 0.333 | 0.552 | 0.972 |
| Chengdu | 0.85 | 0.908 | 0.552 | 0.562 | 0.993 |
| Chongqing | 0.844 | 0.802 | 0.519 | 0.383 | 0.901 |
| Kunming | 0.712 | 0.917 | 0.618 | 0.15 | 0.94 |
| Changzhou | 0.889 | 0.912 | 0.281 | 0.551 | 0.955 |
| Hangzhou | 0.889 | 0.788 | 0.661 | 0.126 | 0.788 |
| Hefei | 0.851 | 0.866 | 0.492 | 0.485 | 0.865 |
| Changsha | 0.937 | 0.694 | 0.273 | 0.663 | 0.671 |
| Nantong | 0.69 | 0.659 | 0.345 | 0.3 | 0.68 |
| Yangzhou | 0.689 | 0.948 | 0.307 | 0.412 | 0.973 |
| Nanchang | 0.776 | 0.845 | 0.558 | 0.122 | 0.829 |
| Shaoxing | 0.681 | 0.836 | 0.477 | 0.144 | 0.933 |
| Guiyang | 0.557 | 0.841 | 0.268 | 0.35 | 0.866 |
| Wuhan | 0.821 | 0.575 | 0.223 | 0.526 | 0.605 |
| Ningbo | 0.755 | 0.32 | 0.36 | 0.54 | 0.292 |
| Taizhou | 0.75 | 0.595 | 0.527 | 0.276 | 0.551 |
| Wenzhou | 0.782 | 0.864 | 0.298 | 0.49 | 0.869 |

Note: $\alpha1$: allometric growth regression coefficient in Equation (5); $\beta1$: current allometric growth index in the model considering the lag effect; $\beta2$: lag term coefficient in the model considering the lag effect; and $R^2$: goodness of fit. The original model and the new model are obtained by referring to formulas (5) and (6), respectively.

Based on the allometric growth model considering the lag effect, this paper studies the impact of stock land and new land on lighting in urban agglomerations. Table 2 shows that the goodness of fit of the new model considering the lag effect is better than that of the original model. The greatest increase is 4.9%, and the average increase is 2.0%. These results indicate that the lag effect between nighttime light and urban land increases among urban agglomerations, and the urban agglomerations

whose β1 values are higher than their β2 values are the Yangtze River Delta, Sichuan-Chongqing and Poyang Lake urban agglomerations. The β2 value is greater than the β1 value only in the case of the Wuhan urban agglomeration. Finally, the β2 value is slightly greater than the β1 value for the Changsha-Zhuzhou-Tan urban agglomeration.

**Table 2.** Allometric growth model results of urban agglomerations considering the lag effect.

| Urban Agglomeration | $\alpha 1$ | Original Model $R^2$ | β1 | β2 | New Model $R^2$ |
|---|---|---|---|---|---|
| Yangtze River Delta urban agglomeration | 1.363 | 0.839 | 0.991 | 0.564 | 0.879 |
| Wuhan city circle | 1.097 | 0.736 | 0.382 | 0.847 | 0.785 |
| Changsha-Zhuzhou-Tan urban agglomeration | 1.07 | 0.765 | 0.651 | 0.539 | 0.769 |
| Poyang Lake urban agglomeration | 0.878 | 0.804 | 0.609 | 0.247 | 0.804 |
| Chengdu-Chongqing urban agglomeration | 1.106 | 0.941 | 0.731 | 0.414 | 0.947 |

Note: α1: allometric growth regression coefficient in Equation (5); β1: current allometric growth index in the model considering the lag effect; β2: lag term coefficient in the model considering the lag effect; and $R^2$: goodness of fit. The original model and the new model refer to Equations (5) and (6), respectively.

## 4.2. Analysis of the Land Intensive Use Level in the YREB

### 4.2.1. The Spatiotemporal Characteristics of Increase in Nighttime Light and the Built-Up Area in the YREB

Based on the increase in total nighttime light and the built-up area in the cities of the YREB from 2001 to 2013 (Figures 6 and 7), we analyze the spatial distributions of their increases before the allometric growth analysis of the land intensive use in the YREB. In Figure 7, the areas with the greatest increase in nighttime light are the southeast coastal cities in the lower reaches of the Yangtze River and Chongqing in the upper reaches of the Yangtze River. Furthermore, the increase in the cities in the lower reaches is on the whole apparently greater than that in other areas, indicating that the economic growth in the region is significant. In general, the increase in nighttime light in the whole YREB gradually decreases from east to west, and the upper and lower reaches greatly differ. However, the core cities, such as Chengdu, Chongqing, Kunming, Wuhan, and Changsha, have relatively large increases in nighttime light. In the inland cities, economic development resources and population tend to gather in the central cities; consequently, there is an increase in the growth in nighttime light in these cities that is greater than that in the surrounding cities. To analyze the increase in nighttime light in different cities, we select the top 20 cities with the greatest increase in nighttime light. We compare the increase in nighttime light every four years (Figure 8), and the increase in nighttime light is the smallest in every city from 2001 to 2005. However, from 2005 to 2009, Shanghai, Ningbo, Nantong, and Jiaxing have definite increases. This period consists of the fastest development years from 2005 to 2009 in these four cities, especially in Shanghai and Ningbo. The growth in urban nighttime light is concentrated from 2009 to 2013 for most cities because coastal cities, such as Shanghai and Ningbo, which have benefited from early policy support and geographical advantages, are the priority cities for China's development. To balance the development of the lower reaches and the development of the middle and upper reaches of the Yangtze River, most of the core cities in the urban agglomeration along the Yangtze River have subsequently obtained high policy support.

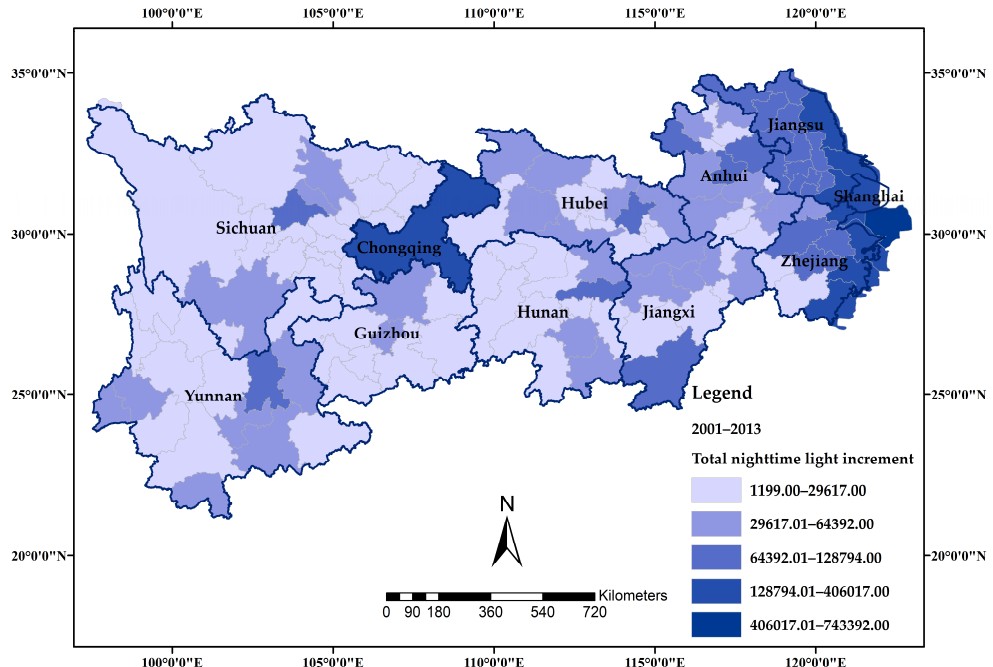

**Figure 6.** Spatial distribution of the increase in total nighttime light in the YREB.

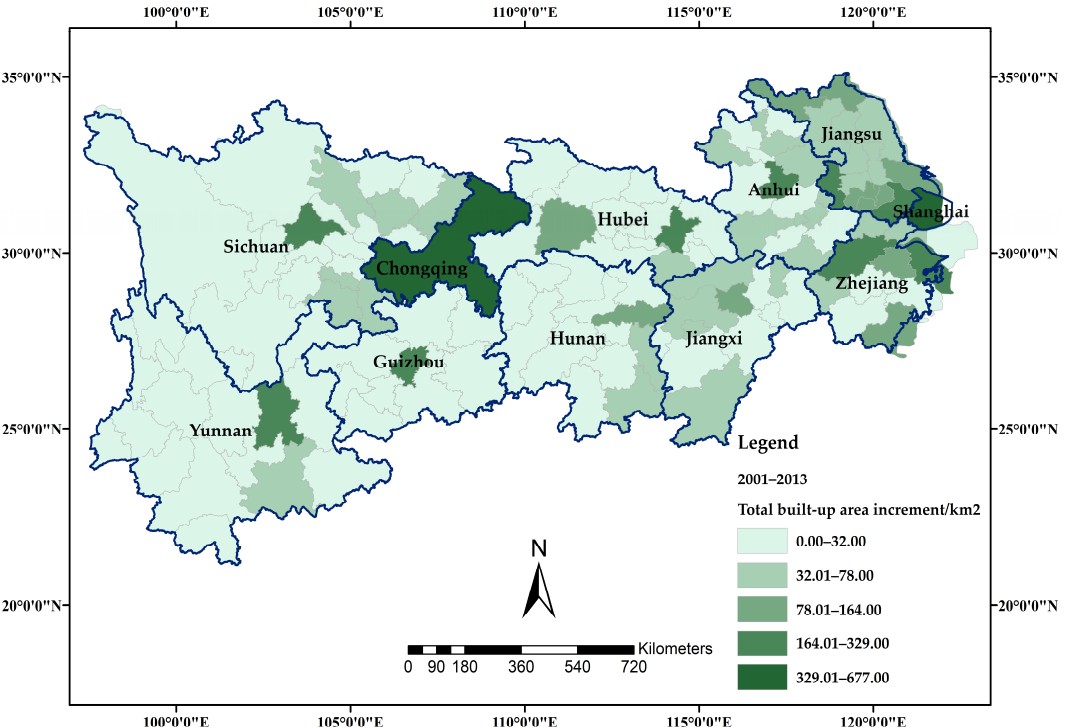

**Figure 7.** Spatial distribution of the increase in total built-up area in the YREB.

Figure 7 shows that the spatial distribution of the increase in built-up area is approximated by that of the increase in nighttime light. The coastal cities in the lower reaches of the Yangtze River and the core cities in the inland area have greater increases in the built-up area, indicating that these regions have high levels of urbanization with large inflows of population. Comparing the growth rates of nighttime light and the built-up area (Figure 8), we find that the growth in nighttime light in all cities is not significant from 2001 to 2005. However, from 2005 to 2009, there is a significant increase. Except for Shanghai and Ningbo, which have the highest increase in nighttime light from 2005 to 2009, the growth in nighttime light in other cities from 2009 to 2013 is higher than that from

2005 to 2009. This result indicates that typical cities have entered a period of rapid urban development since 2005. In addition, they have maintained a good growth trend in the course of nearly ten years of development. In Figure 8b, the growth rates in different cities are not consistent. The growth rate of nighttime light is greater than that of the built-up area in Shanghai, Chongqing, Jiaxing, Hangzhou, Wuxi, Nanjing, Kunming, Changzhou, Jinhua, Changsha, and Shaoxing, indicating that urban land was used efficiently in these cities. However, in Nantong, Suzhou, Chengdu, Wuhan, Yancheng, and Lianyungang, the growth rate of nighttime light is less than that of the built-up area. Suzhou, Chengdu and Lianyungang are the cities with the fastest growth in the built-up area, and it is necessary for them to further explore the value of urban land use. In the following section, we analyze urban land intensive use in the YREB based on the results of a model with nighttime light intensity and the degree of urbanization.

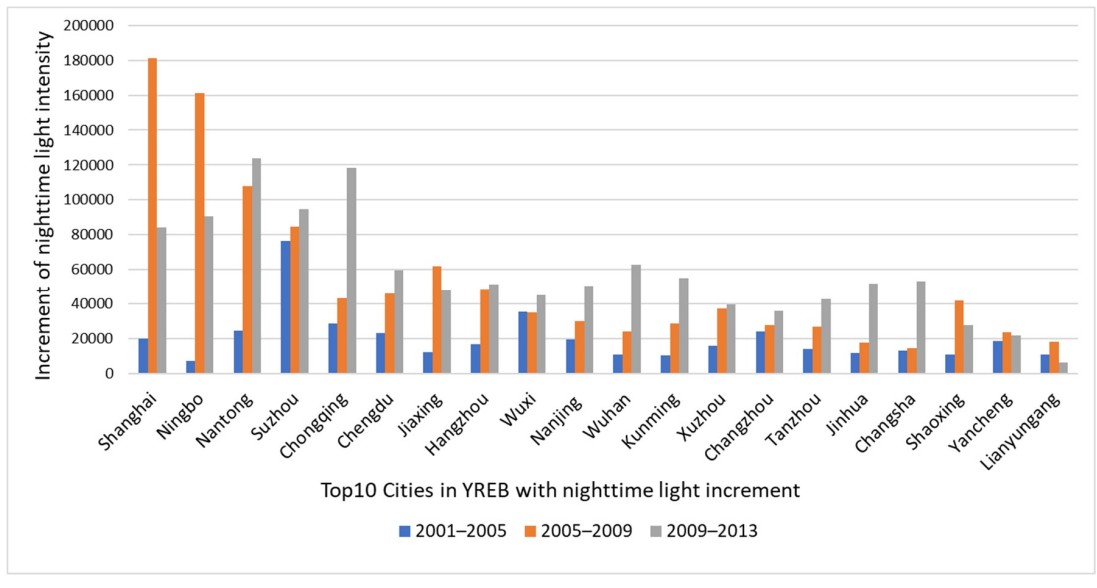

(a)

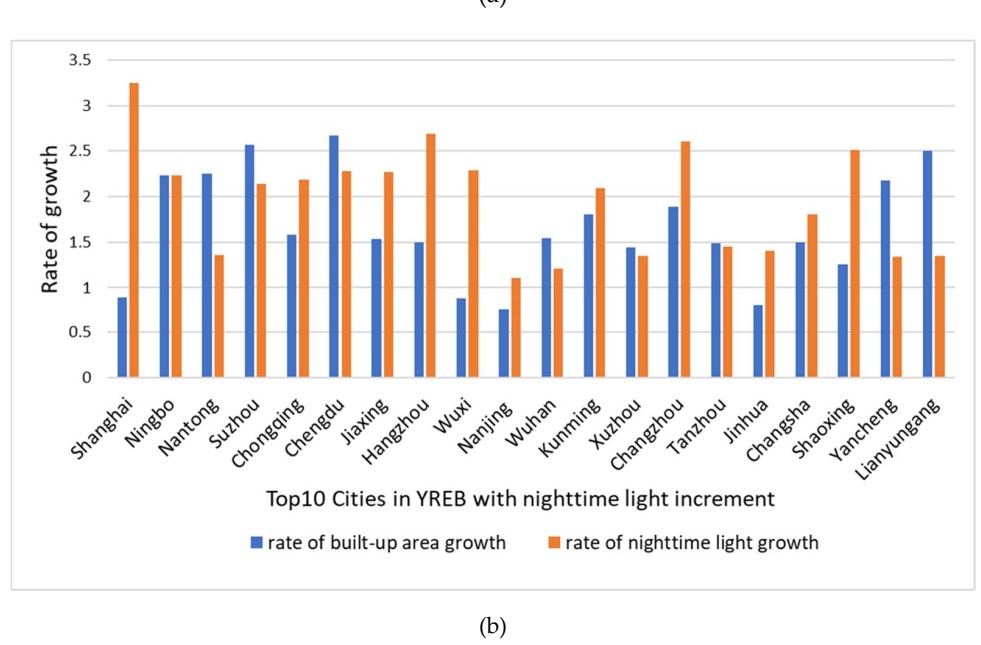

(b)

**Figure 8.** Contrast of growth in nighttime light and the built-up area from 2001 to 2013: (**a**) Increase in nighttime light (2001–2013) every four years; and (**b**) the growth rate of the built-up area and nighttime light.

4.2.2. Analysis of the Spatial Characteristics of the Land Intensive Use Level in the YREB

Figure 9 shows that the proportion of cities with higher levels of land intensive use in the lower reaches of the Yangtze River is much larger than those in the middle and upper reaches of the Yangtze River. In general, the degree of land intensive use in the whole YREB gradually decreases from east to west, and the upper and lower reaches greatly differ.

In the lower reaches of the Yangtze River, the growth rate of nighttime light intensity is faster than that of land expansion in Shanghai, Suzhou, Wuxi and Nanjing over the previous 13 years. The construction land in these cities has a high degree of utilization, population aggregation, economic activity, and shipping centers. Shanghai is the national central city and the national economic, financial, trade and shipping center. The land output efficiency and land intensive use are far higher in Shanghai than in other cities. The urban agglomeration in the Yangtze River Delta with Shanghai as the center has a higher level of land intensity and shows a fan-shaped decreasing pattern, with Shanghai at the center. As the central city of Jiangsu, Nanjing is an important transportation hub, and Suzhou, Wuxi and other economically developed areas in Southern Jiangsu have a high degree of population concentration; thus, the four cities have a high level of intensive land use. The allometric growth coefficients of Changzhou, Taizhou, Hangzhou, Huzhou, Ningbo, and Wenzhou are close to 1. The light intensity in these areas is increasing at nearly the same rate as land expansion, the degree of intensive land use is relatively large, and the urban scale, land development and utilization planning are reasonable. These cities are close to developed cities in the Yangtze River Delta. With the development of urbanization, intercity rail transit in the Yangtze River Delta has rapidly developed. The population mobility among cities is large, and the economic ties are becoming increasingly close. The central developed cities play a very positive role in promoting the economy. However, in most areas of Northern Jiangsu, the allometric growth coefficient is low, the nighttime light intensity shows negative allometric growth relative to urban land expansion, and urban construction land is extensive. In recent years, the population loss in Xuzhou, Yancheng, and Suqian has been serious. Since the scale of northern Jiangsu is insufficient and the proportion of the agricultural industry is larger than that of the manufacturing and service industries, a large number of people flow to southern Jiangsu or Shanghai, and the efficiency of urban land use is low.

The allometric growth coefficients of Wuhan, Jingzhou, and Changsha in the middle reaches of the Yangtze River are lower than those of the eastern developed areas; however, they are clearly superior to those in the surrounding cities. These cities are all in the regional center. The increase in nighttime light intensity is equivalent to the increase in the built-up area. The regional center shows a decreasing trend toward the surrounding areas, and the surrounding urban land use shows moderate and low intensity. According to the "*Development Plan of the Urban Agglomeration in the Middle Reaches of the Yangtze River*" approved by the State Council, the radiation-driving capacity of the central cities of Changsha, Wuhan and Nanchang will be strengthened and will play leading roles in the overall increase in land use efficiency in the middle reaches of the Yangtze River. The land use in the central cities in the middle and lower reaches of the YREB has become saturated. To protect the red line of arable land and avoid an overconcentration of population, the relevant personnel need to balance the allocation of development resources between central cities and surrounding cities to avoid an excessive loss of population in second- and third-tier cities.

However, the upper and lower reaches of the Yangtze River are separated by a gap. Only Chengdu and Chongqing have higher allometric growth coefficients and a higher degree of urban land intensive use. Chengdu is the only subprovincial city and megacity in Southwest China. However, Chengdu's leading role in the development of the surrounding cities is unclear. In policy, Chengdu has an agglomeration effect on the resources and elements of the whole province and the western region, and it is insufficient to drive the development of surrounding cities. In the "*Development Plan of the Chengdu-Chongqing Urban Agglomeration*" approved by the State Council in 2016, the aim is to build Chengdu into a national central city and to radiate outward to promote the development of surrounding cities. The aim is to realize the balanced development of urban agglomeration in

southwest China, with the central cities, mainly Chengdu and Chongqing, driving the surrounding areas. In addition, Yunnan, Guizhou and western Sichuan have large mountainous areas, and the limitation of land resources is also an important reason for the limited economic development and the low level of intensive land use in these areas.

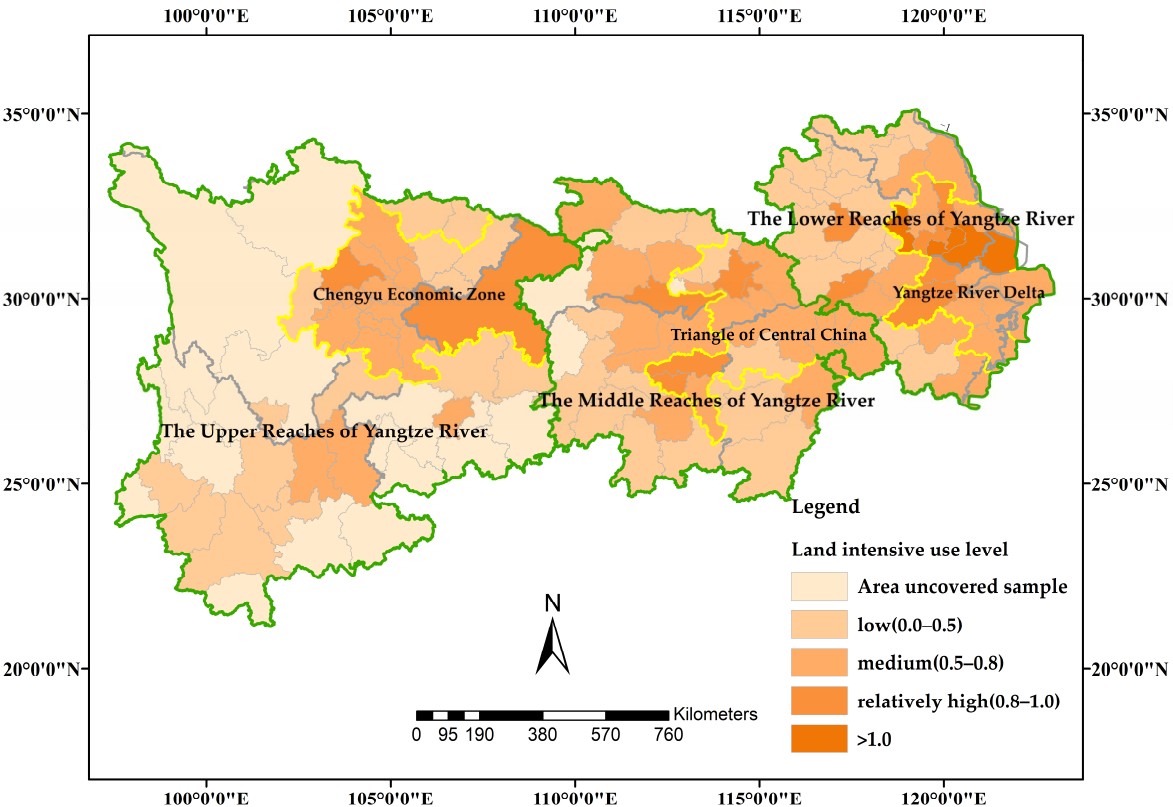

**Figure 9.** Distribution map of land intensive use levels in the YREB.

To analyze and understand the level of land intensive use on a larger scale, this paper selects the Yangtze River Delta urban agglomeration, Wuhan city circle, Changsha-Zhuzhou-Tan urban agglomeration, Poyang Lake urban agglomeration and Chengdu-Chongqing urban agglomeration in the YREB. The Yangtze River Delta and the urban agglomeration of the middle reaches of the Yangtze River are interregional urban agglomerations that form a number of inner-city circles. The center cities of these urban agglomerations have a higher level of land intensive use. As a result of the multicenter structure, the distribution of resources is relatively dispersed. However, the strength of the multicenter cities of the urban agglomeration of the Yangtze River Delta is outstanding, and the efficiency of urban land use is greater than that in the middle reaches. The Chengdu urban agglomeration is still dominated by Chengdu and Chongqing in the upper reaches of the Yangtze River, and the radiation intensity to the surrounding cities is poor, indicating that the development of urban land is spatially concentrated. The result of the urban utilization pattern is consistent with the "*Outline of the Yangtze River Economic Belt Development Plan*" published in 2016. Making full use of the Golden Channel of the Yangtze River to promote the development of underdeveloped areas and to balance the allocation of resources of the eastern and western regions is a major challenge and opportunity. In addition, based on defining the development orientation of each city, realizing the coordination and linkage between the core cities and major cities is the key to enhancing the overall sustainable development ability of the core cities in the YREB. To ensure coordinated and cooperative development among key cities and to realize the radiation of the key cities to surrounding cities, cooperation among cities in the YREB should be broadly and deeply promoted [51].

4.2.3. Analysis of the Lag Effect in Land Intensive Use in the YREB

According to the results for the typical cities based on the model considering the lag effect (Figure 10), we analyze the utilization efficiency of new land and stock land for the level of economic activity during the period of urban area expansion. The impact of stock construction land on the growth in nighttime light in Shanghai, Wuxi, Changzhou, Changsha, Yangzhou, Guiyang, Wuhan, Ningbo, and Wenzhou is greater than the impact of new land. Shanghai, Changsha, and Wuhan, which are also the central cities in the urban agglomeration, are the top three cities with the largest differences between the values of β2 and β1. Although the city scale is different, the utilization efficiency of stock construction land and the degree of urban centralization are high. The population and the main economic activities remain concentrated in the urban center. For these cities, the central land is near saturation, and most of the newly developed land has not flourished in a timely manner. It is necessary to strengthen the utilization of new land. The intensity of new land development is high in Nanjing, Suzhou, Hangzhou, Chongqing, Kunming, Nantong, Nanchang, Shaoxing, and Taizhou. The top three cities with the largest differences between the values of β1 and β2 are Hangzhou, Kunming and Nanjing, which indicates that they have great potential for utilizing the stock of construction land. The utilization of stock urban land needs to be strengthened, and the expansion of new construction land should be reasonably controlled. For Chengdu and Hefei, in the values of β1 and β2 are similar, and the rate of contribution of stock land and new land to nighttime light is balanced, reflecting the development trend of urban integration. Considering the model of typical cities with the lag effect, the intensive land use of typical cities is not affected by the region, and there is no large gap between the cities in the upper and middle reaches and certain cities in the lower reaches with regard to utilization efficiency.

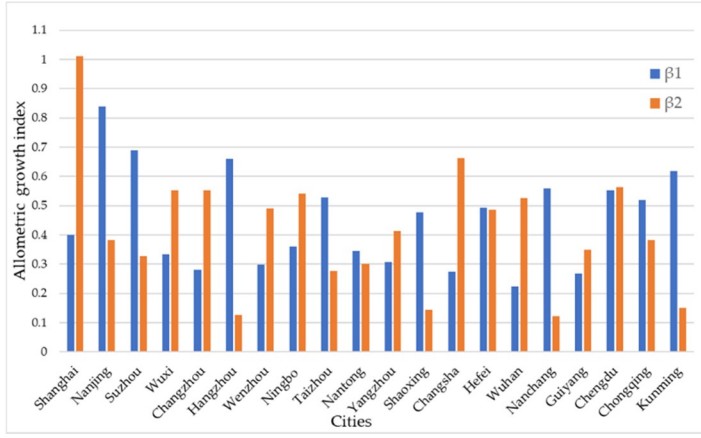

**Figure 10.** Comparison of the coefficients of typical cities based on the model considering the lag effect.

In Figure 11, the urban agglomerations for which the value of β1 is higher than the value of β2 are the Yangtze River Delta, Sichuan-Chongqing, and Poyang Lake urban agglomeration, indicating that new land contributes more to nighttime light in the process of urban expansion, newly expanded urban land can quickly enter the stage of development and utilization, and new land is intensive. Urbanization expansion tends toward radiation. The urbanization process and utilization efficiency of the urban agglomeration in the Yangtze River Delta are much greater than those in the middle and upper reaches. A large number of development zones and open economic development bring a large number of migrants, increasing the efficiency of urban land use. In addition, the utilization ratio of stock construction land is almost saturated; thus, the efficiency of new land development and utilization is high. Unlike the Yangtze River Delta, the land intensive use level in the Chengyu urban agglomeration and the Poyang Lake urban agglomeration is lower, but new land is used more efficiently than stock land. Evidently, the stock land remains underutilized. Future planning should focus on revitalizing the use of the inefficient stock of construction land, especially for the

weak development level around the central city. The value of β2 is greater than the value of β1 for Wuhan city circle, indicating that the growth in nighttime light is mainly concentrated on the stock of land, and Wuhan city circle has a high degree of centralization with regard to cities and towns. Therefore, the efficiency of stock construction land use is high in Wuhan city circle, indicating that urban development tends toward use of the inner space of the urban agglomeration. In the Changsha-Zhuzhou-Tan urban agglomeration, the contributions of previously developed urban land and new land to economic development are basically the same. To realize the sustainable use of land in the YREB, the carrying capacity of the environment and resources should also be evaluated, and the range of suitable development, restricted development and prohibited development areas should be defined in a serious manner [52].

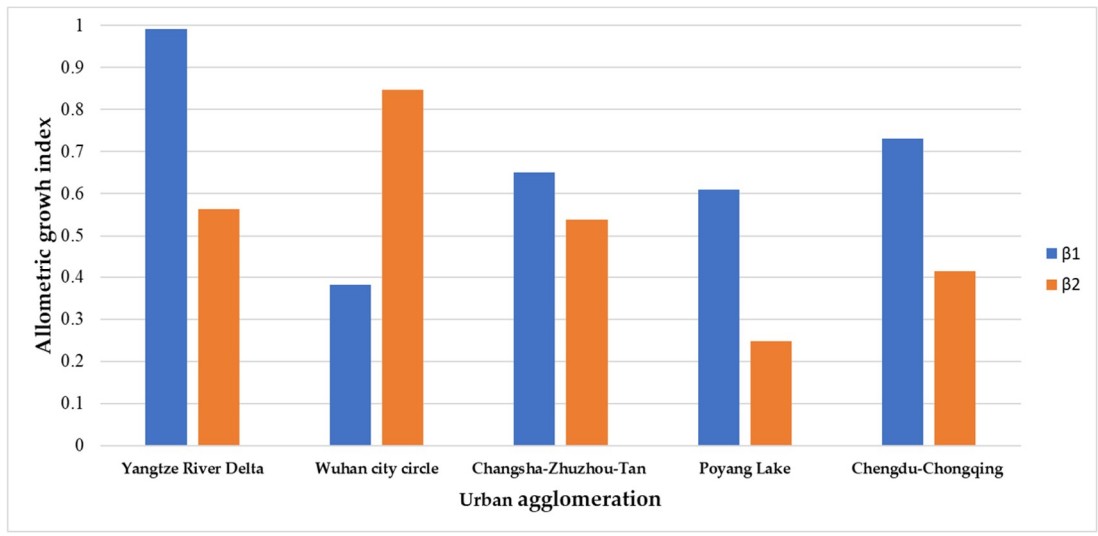

**Figure 11.** Comparison of the urban agglomeration coefficients based on the model considering the lag effect.

## 5. Conclusions

Based on the advantages of long time series nighttime light remote-sensing data in examining social and economic problems, this paper discusses urban land intensive use. Nighttime light intensity and the land urbanization level are used to construct an allometric growth model to analyze the characteristics of urban light and land urbanization in the YREB. Considering the lag of nighttime light growth relative to land development, a new allometric growth model considering the lag effect is used to analyze the level of land intensive use, and the applicability of the model is validated by considering high-population cities and national-level urban agglomerations in the YREB as examples. Nighttime light data can reveal the law of spatiotemporal evolution, and the model that we propose has the ability to mine the rules in different scales of research areas. We draw the following conclusions.

(1) DMSP nighttime light data can effectively compensate for the shortcomings of incomplete traditional counting data, statistical calibration and time-consuming processing, and they positively affect regional research at a large spatiotemporal scale. Nighttime light data lead to good regression results on the urbanization level in the allometric growth model, and they perform well when analyzing urban land intensive use. Nighttime light remote-sensing data are also affected by many complex factors, and because of the low resolution of DMSP/OLS data, they do not perform well in inner-city research. In future research, we could further explore the combination of multiple remote-sensing data sources and some official statistical data to study the forces driving the different levels of land intensive use in the YREB

(2) The allometric growth model can better fit the intensity of urban light and the land urbanization level. The allometric growth coefficient can reflect the land use characteristics of cities and

urban agglomerations. Compared to the original allometric growth model, the goodness of fit of the model considering the lag effect improves to a certain extent. The highest increase in goodness of fit is by 9.9%, and the average increase is by 3.2% in experiments with typical cities. In the experiments based on urban agglomerations, the highest increase in goodness of fit is by 4.9%, and the average increase is by 2%. The model can better reflect the lag effect between growth in nighttime light and urban land expansion, facilitating analysis of the rates of contribution of stock land and new land to the growth in nighttime light. However, this model involves fewer variables. In this research design, no other related variables such as the input and output of different industries were taken into account, which is not conducive to quantitatively and comprehensively analyzing the driving factors. This content also requires further study in the future.

(3)  According to the results, we analyze the urban land use pattern from the perspective of cities of different sizes and the urban agglomerations in the YREB, which reflects the land use rules at different scales of research areas. These rules include the difference and the relationship between the efficiency of new land and developed land. The results show that the degree of intensive land use in the whole YREB gradually decreases from east to west but that the urban agglomerations and cities in the provinces greatly differ. In general, the allometric characteristics of nighttime light and urban land in the lower reaches of the Yangtze River Delta are manifested as positively allometric. The areas with high land use efficiency in the middle and upper reaches are mainly concentrated in the central areas of urban agglomerations, such as Chengdu and Chongqing in the upper reaches and Wuhan and Changsha in the middle reaches of the Yangtze River. New land use contributes more to nighttime light than stock land in the Yangtze River Delta urban agglomeration, the Chengyu urban agglomeration and the Poyang Lake urban agglomeration, indicating that newly expanded urban land can quickly enter the stage of development and utilization and that the utilization potential of stock construction land must be further exploited. For the Yangtze River Delta urban agglomeration, with geographical advantages and early policy support, the intensive use degree is significantly higher than that of other urban areas, and the utilization of stock construction land is almost saturated; thus, the efficiency of new land development and utilization is high. Regarding the Chengyu urban agglomeration and the Poyang Lake urban agglomeration, it is necessary to reasonably control the scale of urban expansion to exploit the value of developed urban land resources. The efficiency of construction land use of Wuhan city center is relatively high, indicating that the urban agglomeration of the city group is highly centralized. To further improve the level of urban land intensive use in the YREB and to realize the sustainable utilization of regional land and sustainable development in an economical manner, provinces and cities must reasonably control the scale of urban land expansion, tap the potential of stock land use for construction, accelerate the coordinated development of different regions, strengthen the links between central cities and the surrounding cities, and strengthen the development radiation intensity of the central cities.

**Author Contributions:** X.C. had the original idea for this study, analyzed the data and wrote the paper. Y.L. and C.S. processed the data; P.L. drew the analysis figures and read the literature. H.S. reviewed the structure of the paper and gave advice on the design of the methodology.

**Funding:** This research was undertaken at the College of Geomatics Science and Technology of Nanjing Tech University and the School of Geography of Nanjing Normal University, and it was funded by the National Natural Science Foundation of China (nos. 41501431, 41601449, and 41771421), as well as the Natural Science Foundation of the Jiangsu Higher Education Institutions of China (no. 16KJD420002).

**Acknowledgments:** We acknowledge the data support from the National Geographic Data Center of the National Oceanic Atmospheric Administration (http://ngdc.noaa.gov/eog/dsmp), the Ministry of Housing and Urban-Rural Development of the People's Republic of China (http://www.mohurd.gov.cn/xytj/tjzljsxytjgb/) and the Yangtze River Delta Science Data Center, National Earth System Science Data Sharing Infrastructure, National Science and Technology Infrastructure of China (http://nnu.geodata.cn).

**Conflicts of Interest:** The authors declare no conflict of interest.

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
