# Peer review of "Urban Land Intensive Use Evaluation Study Based on Nighttime Light—A Case Study of the Yangtze River Economic Belt"

_sustainability, doi:10.3390/su11030675_

Round 1
Reviewer 1 Report
Comments
The topic is interesting and relevant.
The introduction is able to give an exhaustive overview of the state of the art.
The methodology is correct and well written, but the model of equation (1) must be explained with more details.
The discussion of the results obtained is not exhaustive and needed specifications
To complete the conclusions, I suggest to better specify the future prospects for research
Other comments:
The caption of Figure 5a is not correct
The line numbers are missing
In Figure 8, change the type of graph for the rate of build-up and the rate of nighttime light because they are not continuous data
Author Response
Dear Editors and Reviewers:
We would like to express our great appreciation to you and reviewers for comments on our paper. Those comments are all valuable and very helpful for revising and improving our paper, as well as the important guiding significance to our researches. We have studied comments carefully and have made correction which we hope meet with approval. The main corrections in the paper and the responds to the reviewer’s comments are as flowing.
Point1: The methodology is correct and well written, but the model of equation (1) must be explained with more details.
Respond1: The equation (1) shows the relationship between two parts of a system geometrically. x and y are two elements of a system, t means time, and b is the ratio of the growth rate between x and y. When the two correlation measures of a system satisfy equation (1), the system obeys the law of allometric growth. We have given more details for this part at line 203 to line 204.
Point2: The discussion of the results obtained is not exhaustive and needed specifications
Respond2: Our research aims to study the land intensive use level in the YREB. We used nighttime light data and built-up area statistical data to construct model based on the allometric growth model to analyze the urban land intensive use level spatially and temporally. Based on the results of allometric growth model and that with lag term, we discussed the urban land use in YREB from the spatiotemporal characteristic of nighttime light and build-up area, spatial distribution of land intensive use level and the lag effect in urban land use. We have discussed typical cities in different regions based on the results and analyzed the reasons which lead to these results. According to your suggestion, we have detailed the discussion of the results in section4.2.
Point3: To complete the conclusions, I suggest to better specify the future prospects for research
Respond3: In this research, we study the land intensive use in the YREB macroscopically and this model involves two variables. It is not enough to analysis the driving force quantificationally and comprehensively. In the future research, we would combine multiple remote sensing data sources and some official statistical data to analysis the force driving of the different land intensive use level in the YREB providing more favorable support for the development planning of the YREB. According to your suggestion we have added more content about the future prospects in conclusion line563 to line564 and line575 to line577
Point4: The caption of Figure 5a is not correct
Respond4: We are very sorry for our negligence of reverence. We have corrected the caption of Figure 5a.
Point5: The line numbers are missing
Respond5: We are very sorry for our negligence of reverence. We have supplemented the line numbers according to your advice.
Point6: In Figure 8, change the type of graph for the rate of build-up and the rate of nighttime light because they are not continuous data
Respond6: We have divided Figure8 into Fig8(a) and Fig8(b). Fig8(a) shows the increment of nighttime light of these cities every four years and Fig8(b)shows the contrast between rate of built-up area growth and nighttime light growth from 2001 to 2013.

Reviewer 2 Report
Dear authors, I enjoyed reading your paper, it is well written and the research is well designed. To improve the paper I suggest the following, this will make the paper stronger and benefit the readership of this journal more.
Major Comments:
0) Please place line numbers in revised manuscript.
1) It is unclear what the authors mean by land use efficiency. How is this calculated?
Please add a subsection under Methodology and expand on the calculation and interpretation of 'land use efficiency', one or two examples would also be very helpful to the reader.
2) Section 5 first paragraph: "the model that we propose has the ability to mine the rules in different scales of research areas". This is interesting, please expand! Which disciplines?
3) What are the limitations of your research design? and your research results?
4) Figures 10 & 11: What are b1 and b2? Are they related to efficiency? This is somewhat confusing to the reader. Firstly, I suggest to please write down what b1 and b2 are, their formula and description, the first time the authors mention b1 and b2 are in page 9, however, I can't see the related formula, perhaps the authors can enhance this and make it more visible?
5) The paper has really good results but it is weak on the discussion of these results. The authors especially begin the paper with the claim that "The urban land intensive use evaluation of the YREB is helpful for understanding the efficiency of urban land during urban expansion and provides a basis for urbanization in the future. The feasibility of intensive utilization provides a reference for land use efficiency evaluations at
different spatial scales within the YREB." However, I don't see these important points expanded and critically discussed enough. Towards this end, I recommend the authors to:
- I suggest to include a chart on efficiency values of the different cities and interpret the driving forces behind the diverging efficiency values.
- I suggest more discussion on the policy implications of your research. How would this research benefit land use policymakers? What would be their biggest lessons? I think if you expand on these points the paper can be greatly stronger.
- The paper may seem very technical, therefore, to attract a wider audience, I suggest to use more policymaking literature in your discussions, the following may be relevant papers for you to reference: Yarime and Kharrazi “Understanding the Environment as a Complex Natural-Social System: Challenges and Opportunities for Public Policies”, in Bernardo Alves Furtado et. al., ed., Modeling Complex Systems for Public Policies, Brasilia, Brazil. Institute for Applied Economic Research (IPEA), Brasilia, 2015, pp. 127-140.
Minor comments:
Page.Two.Paragraph.One: What is pattern planning? References would be helpful.
Page.Two.Paragraph.One: gradually expanding
Page.Two.Paragraph.Three: Very interesting, this might be a digression but the Kardashev scale also evaluates the maturity of a civilization by the amount of energy it uses. I am curious if there is a similar scale based on the urban nightlight.
Figure 5 is difficult to read, I would suggest the authors use bigger and bolder fonts.
Author Response
Dear Editors and Reviewers:
We would like to express our great appreciation to you and reviewers for comments on our paper. Those comments are all valuable and very helpful for revising and improving our paper, as well as the important guiding significance to our researches. We have studied comments carefully and have made correction which we hope meet with approval. The main corrections in the paper and the responds to the reviewer’s comments are as flowing.
Point1: Please place line numbers in revised manuscript.
Response1: We are very sorry for our negligence of reverence. The revised manuscript has been placed line numbers according to your suggestion.
Point2: It is unclear what the authors mean by land use efficiency. How is this calculated?
Please add a subsection under Methodology and expand on the calculation and interpretation of 'land use efficiency', one or two examples would also be very helpful to the reader.
Response2: Urban land use refers to the process of land development by land owners or users according to certain economic purposes and functions of land allocation. In our research, we focus on the urban land use efficiency. Generally, if limited developed urban land resource produces considerable economic benefits, the urban land use efficiency is high. Urban land use efficiency can be measured by population density, volume rate and rate of land output in cities based on some references. We added the content in the introduction at line76 to line79.
Because nighttime light has the unique ability to reflect human social activities and economic development. The urbanization enables to influence the growth of nighttime light. Therefore, the nighttime light intensity can reflect whether the developed land is utilized efficiently. Allometric growth coefficients are used to determine the allometric growth characteristic, which includes positive allometric, isometric allometric and negative allometric respectively. If the allometric growth characteristic is positive or isometric in a city, we regard the land use is efficient in this city. Considering the reviewer’s suggestions, we added the interpretation of land use efficiency in introduction with some references. The statement has been added at line227 to line236.
Point3: Section 5 first paragraph: "the model that we propose has the ability to mine the rules in different scales of research areas". This is interesting, please expand! Which disciplines?
Respond3: In this research, we choose cities and urban agglomerations in YREB as our research objects. The ability to mine the rules means that this method reveals the regular pattern during the development of YREB. And we analysis the urban land use pattern from the perspective of cities in different size and urban agglomerations, which reflects the land use rules in different scales of research areas. The rules include the difference of land use efficiency and the relationship between new land and developed land in time series among different areas. We have added more statement to expand this view in conclusion point3 at line578 to line581.
Point4 What are the limitations of your research design? and your research results?
Response4: Because this method involves fewer variable, no more other related variables were taken into account, so it is not conducive to driving force analysis comprehensively. And the limitation of the research results is that we did not analysis driving force quantificationally. We will do more research in the future. We have supplemented the statement in conclusion point2 at line574 to line577.
Point5: Figures 10 & 11: What are b1 and b2? Are they related to efficiency? This is somewhat confusing to the reader. Firstly, I suggest to please write down what b1 and b2 are, their formula and description, the first time the authors mention b1 and b2 are in page 9, however, I can't see the related formula, perhaps the authors can enhance this and make it more visible?
Reponse5: We are sorry for our negligence of reverence. b1 and b2 are and respectively in formula (5). is the allometric growth coefficient of the current period and is the lag term coefficient. We have unified the character and the whole manuscript uses and .
Point6: The paper has really good results but it is weak on the discussion of these results. The authors especially begin the paper with the claim that "The urban land intensive use evaluation of the YREB is helpful for understanding the efficiency of urban land during urban expansion and provides a basis for urbanization in the future. The feasibility of intensive utilization provides a reference for land use efficiency evaluations at different spatial scales within the YREB." However, I don't see these important points expanded and critically discussed enough. Towards this end, I recommend the authors to:
- I suggest to include a chart on efficiency values of the different cities and interpret the driving forces behind the diverging efficiency values.
- I suggest more discussion on the policy implications of your research. How would this research benefit land use policymakers? What would be their biggest lessons? I think if you expand on these points the paper can be greatly stronger.
- The paper may seem very technical, therefore, to attract a wider audience, I suggest to use more policymaking literature in your discussions, the following may be relevant papers for you to reference: Yarime and Kharrazi “Understanding the Environment as a Complex Natural-Social System: Challenges and Opportunities for Public Policies”, in Bernardo Alves Furtado et. al., ed., Modeling Complex Systems for Public Policies, Brasilia, Brazil. Institute for Applied Economic Research (IPEA), Brasilia, 2015, pp. 127-140.
Response6:
For the first suggestion, we use land intensive use level calculated by build-up area and nighttime light data based on allometric growth model to represent the land use efficiency. The allometric growth coefficient is used to assess the land use efficiency. However, because this method involves only two variables, it is difficult to interpret the driving forces quantificationally. And because the number of cities is very large, it’s difficult to show all the land intensive use level values of different cities. Therefore, we used distribution map to show the difference of land use efficiency in different cities within YREB. And then we selected 20 typical cities to compare the efficiency values in Table1 and Figure10. We stated the reasons for different land use efficiency from geography, economy, policy and other factors in section 4.2.
For the second suggestion, we have expanded on policy implications and benefit to land use policymakers in section4.2 and conclusion according to your advice. Firstly, it, is a big challenge and opportunity to make full use of the Golden Channel of Yangtze River in order to promote the sub-developed areas, balance the allocation of resources of the eastern and western regions. Secondly, land use in central cities has become saturated. In order to protect the red line of arable land and avoid over-concentration of population, relevant personnel need to balance the allocation of development resources between central cities and surrounding cities to avoid over-loss of population in second- and third-tier cities. (line456-459)Thirdly, for some cities, the central land is near saturation, most of the newly developed land has not vitalized in time. It is necessary to strength the utilization of new land(line506-507). And for another cities whose stock urban land still has potential to excavate needs to control the expansion of new construction(line511-512).
For the third suggestion, thank you for the relevant papers you mentioned firstly. We have added some policymaking literatures to improve the readability of discussions.(line487-494 and line537-540)
50.Sun, Y.N. The evaluation about the sustainability of core cities in the Yangtze River Economic Belt. Nanjing Journal of Social Sciences 2016, 8, 151-156. doi: 10.15937/j.cnki.issn1001-8263.2016.08.024
51. Cen, X. Y.; Zhou, Y. K.; Shan, W. The resources and environment pattern and sustainable development of Yangtze River Economic Zone. China Development 2015, 3, 1-9. doi: 10.15885/j.cnki.cn11-4683/z.2015.03.001
Point7: Page.Two.Paragraph.One: What is pattern planning? References would be helpful.
Respond7: new development pattern planning is “one axis, two wings, three poles and multi-point” in the YREB which is issued by The Outline of the Development Planning of the YREB in 2016. “One axis” is the golden waterway of the Yangtze River. Shanghai, Wuhan and Chongqing play the important role in promoting the economic development from coastal to inland. “Two wings” refers to the two major transport channels of Shanghai-Chengdu and Shanghai-Ruijin. “Three poles” refers to the Yangtze River Delta urban agglomeration, the middle reaches of the Yangtze River urban agglomeration and Chengdu-Chongqing urban agglomeration. “Multi-point” refers to giving full play to the supporting role of cities outside the three major urban agglomerations.(line52-58)
Point8: Page.Two.Paragraph.One: gradually expanding
Respond8: We are sorry for our negligence of reverence. We have corrected the grammatical error.
Point9: Page.Two.Paragraph.Three: Very interesting, this might be a digression but the Kardashev scale also evaluates the maturity of a civilization by the amount of energy it uses. I am curious if there is a similar scale based on the urban nightlight.
Respond9: Thank you to mention this scale that we are not familiar with. And we got to know about it seriously. The utilization of nighttime light for socio-economic assessment is at the beginning of the trial. It has not risen to the point where similar scale can be summarized.
Point9: Figure 5 is difficult to read, I would suggest the authors use bigger and bolder fonts.
Respond9: We are very sorry for the inconvenience in your reading. We have improved the figures more clearly.

Reviewer 3 Report
The paper presents the results of an interesting research. However, they are many aspects that must to be improved. First, I found the methodology unclear and with many shortcomings. The description of allometric growth is not original. Second, the references are too much oriented on Chinese experience. Only a short look to international literature connected with the topic and I found many results (Butt (2012), Xintong Li, Xinran Wang, Jiang Zhang & Lingfei Wu (2015) in Nature, Gradinaru et al, (2017) in Sustainability, Artmann et al (2019) in Ecological Indicators). The discussion and conclusions are weak and must to be enhanced. Also, the authors have to add the limitations of the methods.
Author Response
Dear Editors and Reviewers:
We would like to express our great appreciation to you and reviewers for comments on our paper. Those comments are all valuable and very helpful for revising and improving our paper, as well as the important guiding significance to our researches. We have studied comments carefully and have made correction which we hope meet with approval. The main corrections in the paper and the responds to the reviewer’s comments are as flowing.
Point1: First, I found the methodology unclear and with many shortcomings. The description of allometric growth is not original.
Respond1: The allometric growth model is not original indeed. This model is used to describe the nonlinear relationship between urban population and urban areas in some related reference. We considered that growth of nighttime light and built-up area may have allometric growth characteristic. The model is used to describe the relationship between the growth rate of nighttime light and one of urban expansion firstly. After the experiment, we found urban nighttime light and urbanization level conform to the characteristic and the allometric growth coefficient performed well in the assessment of land intensive use level. However, in this model, the nighttime light intensity of each year only corresponds to the value of urbanization level in the same year. It only considers the contribution of current urban area increase to nighttime light. Then considering the lag effect of the nighttime light, we added the lag term to the model. In the new model, the nighttime light intensity of each year corresponds to the value of urbanization level both in the same year and last year. This is the most important improvement in this research. And results showed that it is helpful to evaluate the urban land intensive use in cities and urban agglomeration. As a result, this method has achieved our purpose on the urban land intensive use in the YREB well.
Point2: the references are too much oriented on Chinese experience. Only a short look to international literature connected with the topic and I found many results (Butt (2012), Xintong Li, Xinran Wang, Jiang Zhang & Lingfei Wu (2015) in Nature, Gradinaru et al, (2017) in Sustainability, Artmann et al (2019) in Ecological Indicators).
Respond2: We have added some international literatures connected with the topic according to your suggestion.
8.Xiao, W.; Wei, Q. Q. Intensive Land Use Evaluation of Urban Development Zones: A Case Study of Xi’an National Hi-Tech Industrial Development Zone in China. Modeling Risk Management in Sustainable Construction 2010. 245-250. doi: 10.1007/978-3-642-15243-6_28
This paper chooses 16 indexes according to Intensive Land Use Evaluation Protocols of Development Zones to evaluate the intensive land use level. And the weights of indexes are determined by using Delphi method.
17.Simona, R. G.; Christian, L.L.; Lleana, P. S. Are Spatial Planning Objectives Reflected in the Evolution of Urban Landscape Patterns? A Framework for the Evaluation of Spatial Planning Outcomes. Sustainability 2017,9, 1279. doi: 10.3390/su9081279
They aim to assess the implementation of national spatial planning objectives in urban landscapes through the use of an evaluation framework, which makes use of spatially explicit information.
33.Ting, M.; Zhou, Y.K.; Zhou, C.H. Night-time light derived estimation of spatio-temporal characteristics of urbanization dynamics using DMSP/OLS satellite data. Remote Sensing of Environment 2015, 158, 453-464.doi: 10.1016/j.rse.2014.11.022
The DMSP/OLS nighttime light satellite data is used to estimate the spatio-temporal characteristic of urbanization dynamics in this research which proves that nighttime light data perform well in urbanization dynamics.
Point3: The discussion and conclusions are weak and must to be enhanced. Also, the authors have to add the limitations of the methods.
Respond3: Our research aims to study the land intensive use level in the YREB. We used nighttime light data and built-up area statistical data to construct model based on the allometric growth model to analyze the urban land intensive use level spatially and temporally. Based on the results of allometric growth model and that with lag term, we discussed the urban land use in YREB from the spatiotemporal characteristic of nighttime light and build-up area, spatial distribution of land intensive use level and the lag effect in urban land use. We have discussed typical cities in different regions based on the results and analyzed the reasons which lead to these results. According to your suggestion, we have expanded the discussion and conclusion according to your advice.
As for the limitations of the methods, because this method involves fewer variable, no more other related variables were taken into account, so it is not conducive to driving force analysis comprehensively. And the limitation of the research results is that we did not analysis driving force quantificationally. We will do more research in the future. We have supplemented the statement in conclusion point2.

Round 2
Reviewer 2 Report
Good effort, the paper has improved overall.
Two recommendations for the authors:
- Your references are very narrowly focused on Chinese academia, it would be encouraged to diversify your reference resources. This may in fact lead to more citations of your work.
- Some of the revisions need to be edited for grammar and flow.
Author Response
Point1: Your references are very narrowly focused on Chinese academia, it would be encouraged to diversify your reference resources. This may in fact lead to more citations of your work.
Response1: We have added some international literatures about international experiences connected with the topic according to your suggestion.
1. Martina, A.; Luis, I.; Fan P.P. Urban sprawl, compact urban development and green cities. How much do we know, how much do we agree? Ecological Indicators 2019,96,3-9, doi:10.1016/j.ecolind.2018.10.059
2. Hashem, D.; Fardis, S. Urban sprawl on natural lands: analyzing and predicting the trend of land use changes and sprawl in Mazandaran city region, Iran. Environment, Development and Sustainability 2018, 1-22, doi: 10.1007/s10668-018-0211-2
3. Oriol,N.; Joan,L.; Jordi, M. Energy and urban form. The growth of European cities on the basis of night-time brightness. Land Use Policy 2017,61,103-112, doi:10.1016/j.landusepol.2016.11.007
Point2: Some of the revisions need to be edited for grammar and flow.
Respond2: We have checked the grammar and flow carefully and corrected errors according to your suggestion.
Special thanks to you for your good comments.

Reviewer 3 Report
The text need text editing (including a English proofing).
Author Response
Point1: The text need text editing (including an English proofing)
Response1: We try our best to edit the text carefully, and I have used a professional English editing service by American Journal Experts to improve the English expression.
